# Tumor microbiome contributes to an aggressive phenotype in the basal-like subtype of pancreatic cancer

Wei Guo [1,2,5], Yuchao Zhang[2,3,5], Shiwei Guo[4,5], Zi Mei[2], Huiping Liao[2], Hang Dong[2], Kai Wu[2], Haocheng Ye[2], Yuhang Zhang[2], Yufei Zhu[2], Jingyu Lang [2], Landian Hu [2✉], Gang Jin [4✉] & Xiangyin Kong [1,2✉]

Despite the uniform mortality in pancreatic adenocarcinoma (PDAC), clinical disease heterogeneity exists with limited genomic differences. A highly aggressive tumor subtype termed 'basal-like' was identified to show worse outcomes and higher inflammatory responses. Here, we focus on the microbial effect in PDAC progression and present a comprehensive analysis of the tumor microbiome in different PDAC subtypes with resectable tumors using metagenomic sequencing. We found distinctive microbial communities in basal-like tumors and identified an increasing abundance of *Acinetobacter*, *Pseudomonas* and *Sphingopyxis* to be highly associated with carcinogenesis. Functional characterization of microbial genes suggested the potential to induce pathogen-related inflammation. Host-microbiota interplay analysis provided new insights into the tumorigenic role of specific microbiome compositions and demonstrated the influence of host genetics in shaping the tumor microbiome. Taken together, these findings indicated that the tumor microbiome is closely related to PDAC oncogenesis and the induction of inflammation. Additionally, our data revealed the microbial basis of PDAC heterogeneity and proved the predictive value of the microbiome, which will contribute to the intervention and treatment of disease.

[1] Key Laboratory of Stem Cell Biology, Shanghai Jiao Tong University School of Medicine (SJTUSM) & Shanghai Institute of Nutrition and Health, Chinese Academy of Sciences (CAS), Shanghai, China. [2] CAS Key Laboratory of Tissue Microenvironment and Tumor, Shanghai Institute of Nutrition and Health, Chinese Academy of Sciences, Shanghai, China. [3] School of Life Science, Fudan University, Shanghai, China. [4] Department of General Surgery, Changhai Hospital, Second Military Medical University, Shanghai, China. [5] These authors contributed equally: Wei Guo, Yuchao Zhang, Shiwei Guo.
✉email: ldhu2013@163.com; jingang@smmu.edu.cn; xykong@sibs.ac.cn

Pancreatic ductal adenocarcinoma (PDAC) is the most common type of pancreatic cancer and the third most lethal cancer worldwide[1]. Most patients with PDAC have a dismal prognosis, with a 5-year overall survival of only 9%. Surgical resection remains the main option for PDAC treatment; however, the recurrence rate is relatively high, and most patients will eventually die due to metastasis[2,3]. Despite uniform mortality, clinical heterogeneity among patients has been discovered. Among patients with surgically resectable tumors, some progress to an advanced stage within months, while others stabilize for long-term survival[4,5]. A molecular subtype of PDAC termed 'basal-like', which has worse outcomes and a more aggressive phenotype, was identified by transcriptome analysis in a recent publication[6]. Many genomic studies have attempted to elucidate the mechanism of distinct disease progression; however, limited mutational variations have been found[7,8]. The cause of the clinical heterogeneity of tumors remains unclear.

In the past two decades, pancreatic cancer has been recognized as an inflammation-driven cancer in which patients suffering from chronic pancreatitis carry a 13-fold higher risk of PDAC development[9]. Both innate and adaptive immune cell subsets cooperate through various mechanisms to promote tumorigenesis[10]. Consistently, an activated inflammatory stromal response has been considered responsible for PDAC progression[11]. However, the precise mechanisms underlying the pathogenic role of inflammation are still unknown. Increasing evidence suggests that the human microbiome is likely to play a role in activating immune receptors and inducing cancer-associated inflammation[12]. Past studies focusing on the gut microbiome have confirmed the critical effect in intestinal tract malignancies, including esophageal, gastric and colorectal cancers[13]. Recently, the presence of intratumor bacteria in human PDAC has been confirmed[14]. The presence of *Gammaproteobacteria* in pancreas was proven to modulate tumor resistance to gemcitabine. The diverse microbiota alterations have been found in patients with PDAC compared to healthy subjects at oral, gastrointestinal and intrapancreatic tissues[15]. Many studies have found that oral or gut microbiota can translocate to pancreas and play a pivotal role in pancreatic carcinogenesis through several pathways including inflammation, immunity, metabolism, and hormonal regulation[16]. Pushalkar et al. reported a 1000-fold increase of intrapancreatic bacteria in PDAC and demonstrated that the tumor microbiome promotes oncogenesis by inducing immune suppression[17]. Moreover, Riquelme et al. demonstrated that tumor microbiome composition is able to influence pancreatic cancer outcomes by activating antitumor immune response[18]. The intratumor bacteria in pancreatic cancer were also confirmed by Nejman et al., and they found that bacteria are mostly intracellular and are present in both cancer and immune cells[19]. Besides, the fungal mycobiome was shown to be implicated in the pathogenesis of PDAC via activation of MBL by Aykut et al. study[20]. The essential role of the microbiome in immune regulation and oncogenesis was highlighted in recent years that modulations of the microbiome can be profound for the immunotherapy against pancreatic cancer[21]. These findings suggest the implication of the pancreatic microbiome in tumorigenesis, inspiring us to investigate the effect of the microbiome on the clinical heterogeneity of tumor subtypes.

The microbiome composition in human PDAC remains incompletely studied, and whether it produces a favorable or adverse contribution to disease progression demands further exploration. In this study, we presented a comprehensive analysis of the tumor microbiome in different molecular subtypes from 62 resected PDAC. In addition, we performed functional characterization of microbial genes to elucidate the potential of a specific microbiome in inducing inflammation. Our data demonstrated the host-microbiota interplay and found a close association between the microbiome and tumor progression. We also discovered the influence of host genetics in shaping the tumor microbiome. Overall, our work shows that specific microbial compositions promote pancreatic cancer and contribute to aggressive phenotypes by inducing inflammation.

## Results

**Basal-like tumors of PDAC demonstrate activated antimicrobial immunity and inflammatory response**. The molecular classification of pancreatic cancer has been confirmed by several studies[6,22–24]. Different tumor subtypes exhibit distinctive prognoses, which motivated us to explore the cause of tumor-specific progression. Following the subtyping scheme by Chan-Seng-Yue et al.[23], we performed tumor subtype clustering using published gene signatures on our cohort. For the 62 PDAC samples in the cohort, clear clustering was revealed by 4 groups of tumor-related signatures (Fig. 1a). The CDF demonstrated that $k = 2$ underfits the dataset, whereas $k = 4$ or 5 overfits it (Fig. 1b). Therefore, we chose $k = 3$ as the optimal number of clusters, and three main tumor subtypes were identified in our cohort. Signatures 2 and 10 were defined as 'basal-like' signatures by Michelle et al., while genes from signatures 1 and 6 corresponded to the 'classical' program described by both Michelle et al. and Moffitt et al.[6,23]. Based on this evidence, the three subtypes were labeled 'basal-like', 'classical' and 'hybrid'. To validate our clustering, we applied the subtyping model by Moffitt et al. to align the classification, and almost identical subtyping results were obtained (Fig. 1c). We conducted the survival analysis to compare the difference between basal-like and classical tumors, and found a clear tendency of worse survival in basal-like tumors (Supplementary Fig. 1). This finding was consistent with previous studies that have demonstrated the poor outcomes of basal-like subtype[6,23]. By comparing gene expression in the basal-like versus classical subtypes, we identified 2100 genes that showed significant differences, among which 679 genes were upregulated in the basal-like subtype (Fig. 1d). GSEA demonstrated that our basal-like subtype was enriched for terms related to DNA replication (E2F targets, G2/M checkpoint and MYC targets), TGF-β signaling, epithelial mesenchymal transition (EMT) and inflammatory response (Fig. 1e). These findings indicated a more aggressive phenotype of basal-like tumors, which is consistent with previous publications[11,23]. Given that increased TNFα and interferon-γ response represent an activated antimicrobial immune status, we next wondered whether immune infiltration differs between tumor subtypes. We assessed immune cell distribution using CIBERSORTx for each sample and found that the levels of memory B cells, follicular helper T cells, and activated mast cells were significantly elevated in basal-like tumors (Fig. 1f). These data suggested that the aggressive progression of basal-like tumors may be attributed to an excessive immune reaction and inflammation induced by pathogens.

**Basal-like tumors harbor distinctive microbial communities**. Increasing evidence has highlighted that the gut microbiota plays a key role in the activation of the immune system and promoting cancer-associated inflammation[25,26]. To explore the role of the tumor microbiome in the progression of PDAC, we conducted taxonomic assignment using metagenomic sequencing data to build the microbial abundance profile (Supplementary Data 1). A total of 365 distinct microbial genera were identified in our cohort, and the predominant microbiota composition landscape was depicted with the relative abundance of the top 15 taxa (Fig. 2a, b). Tumors of different subtypes harbored a similar component of predominant microorganisms; however, there were

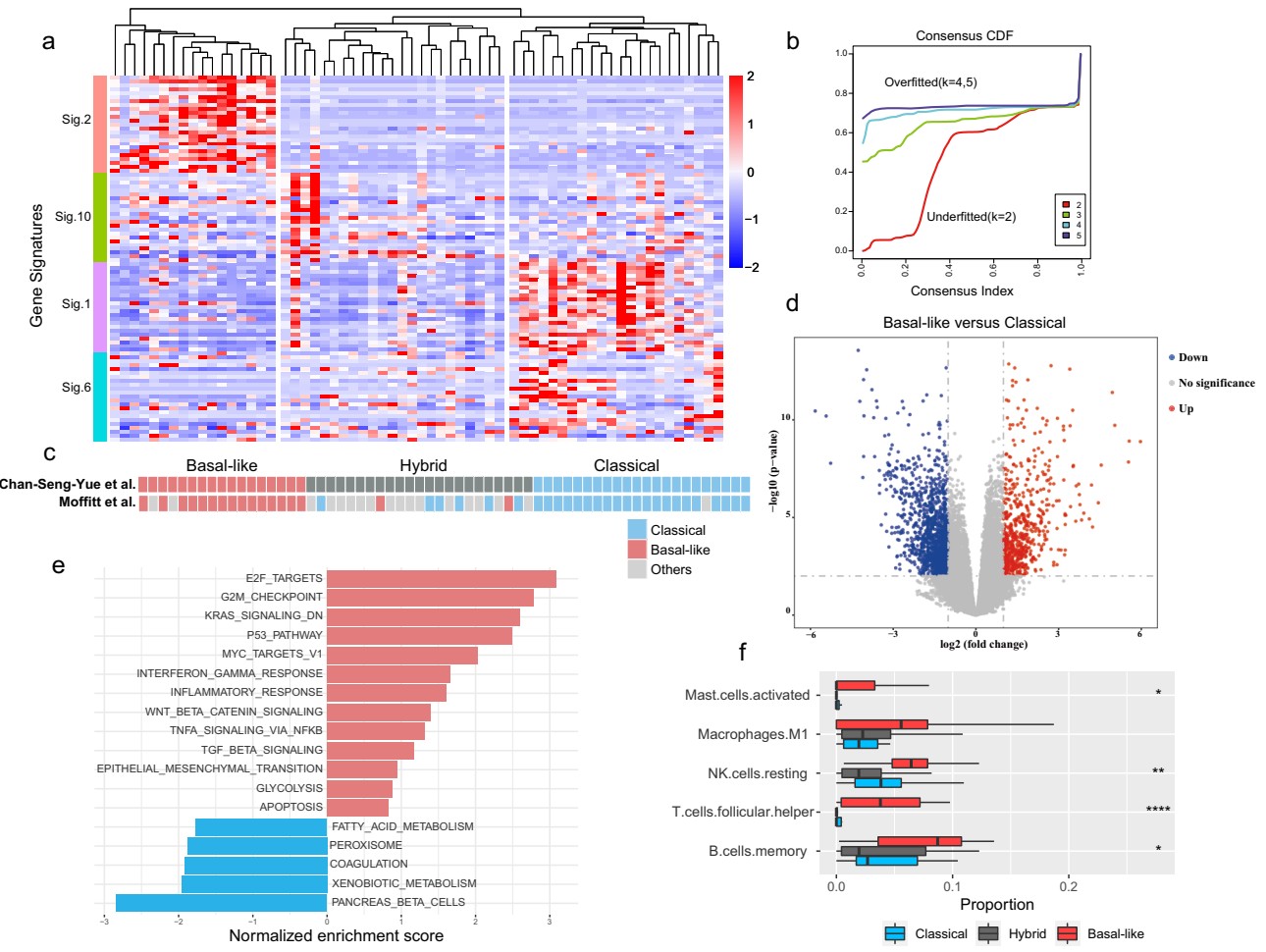

**Fig. 1 Identification of PDAC subtypes by gene signatures. a** Heatmap of the three consensus clusters based on 4 groups of tumor-related signatures determined by Chan-Seng-Yue et al.[23]. Expression values were scaled in the row direction. Signatures 2 and 10 are basal-related signatures, while signatures 1 and 6 are classical-related signatures. Our PDAC samples were classified into basal-like ($n = 17$), hybrid ($n = 23$) and classical ($n = 22$) tumors. **b** Cumulative distribution function (CDF) plot from consensus clustering for different $k$ values in our RNA-seq cohort. We chose $k = 3$ as the optimal cluster number. **c** Comparison of tumor clusters in (**a**) to the previous subtyping scheme by Moffitt et al.[6]. **d** Volcano plot showing differential gene expression in basal-like tumors compared with classical tumors. The up- and down-regulated genes were determined by adjusted $P$ value < 0.01 and log$_2$ fold change >1 and <1, respectively. **e** Enriched pathways of differentially expressed genes identified by GSEA in basal-like versus classical tumors. The length of bars denotes the normalized enrichment score of each category. **f** The boxplot denotes the proportion of five selected types of immune cells among basal-like, hybrid and classical tumors by CIBERSORTx. $P$ values were determined by the Kruskal–Wallis test. * indicates the significance level.

differences in abundance. The top genus, *Pseudomonas*, which is related to infections and has been demonstrated to be associated with short-term survival in PDAC patients[18], showed an increasing abundance in basal-like tumors. To extend the investigation of microbial compositions among tumor subtypes, we measured the tumor microbial diversity using richness (the number of observed taxonomic units), Shannon index of alpha-diversity, and Bray-Curtis metric distance of beta-diversity. As shown in Fig. 2c, the richness of the microbiome was significantly higher in basal-like tumors than in classical or hybrid tumors ($p = 0.0017$ and $p = 0.0026$), while the Shannon index was significantly decreased in basal-like tumors relative to that in classical tumors ($p = 0.017$). Increased richness with a decreased Shannon index indicated a microbial community with diverse members but dominated by a few species with extremely high abundance. We also observed that the Bray–Curtis dissimilarity within basal-like tumors was much higher than that within classical or hybrid tumors. A clear clustering between tumor subtypes (basal-like versus classical/hybrid) was revealed by PCoA, suggesting that the tumor microbiome showed

phylogenetic closeness within each tumor subtype (Fig. 2d). Microbial abundance profiles at the species, family, order and class levels were also built in this study (Supplementary Fig. 2). The diversity measures at various taxonomic levels showed similar tendencies across the three tumor subtypes. Based on these data, we inferred that the tumor microbiome may in part contribute to the specific progression of PDAC subtypes.

**Basal-like subtypes of PDAC show a significant enrichment of tumor microbiome**. The relationship between tumor microbial compositions and PDAC subtypes inspired us to further explore the tumor-related microorganisms. To this end, we analyzed the enrichment of the microbiome in each tumor subtype at various taxonomic levels (Supplementary Data 2). We focused on the microbial features at the genus level because the lower taxonomic level of species showed relatively ambiguous results in estimating accurate abundance. Interestingly, the microbiome in classical and hybrid tumors were similar; however, basal-like tumors harbored quite discriminative microbial communities. This finding was consistent with our previous result in Fig. 2d. We

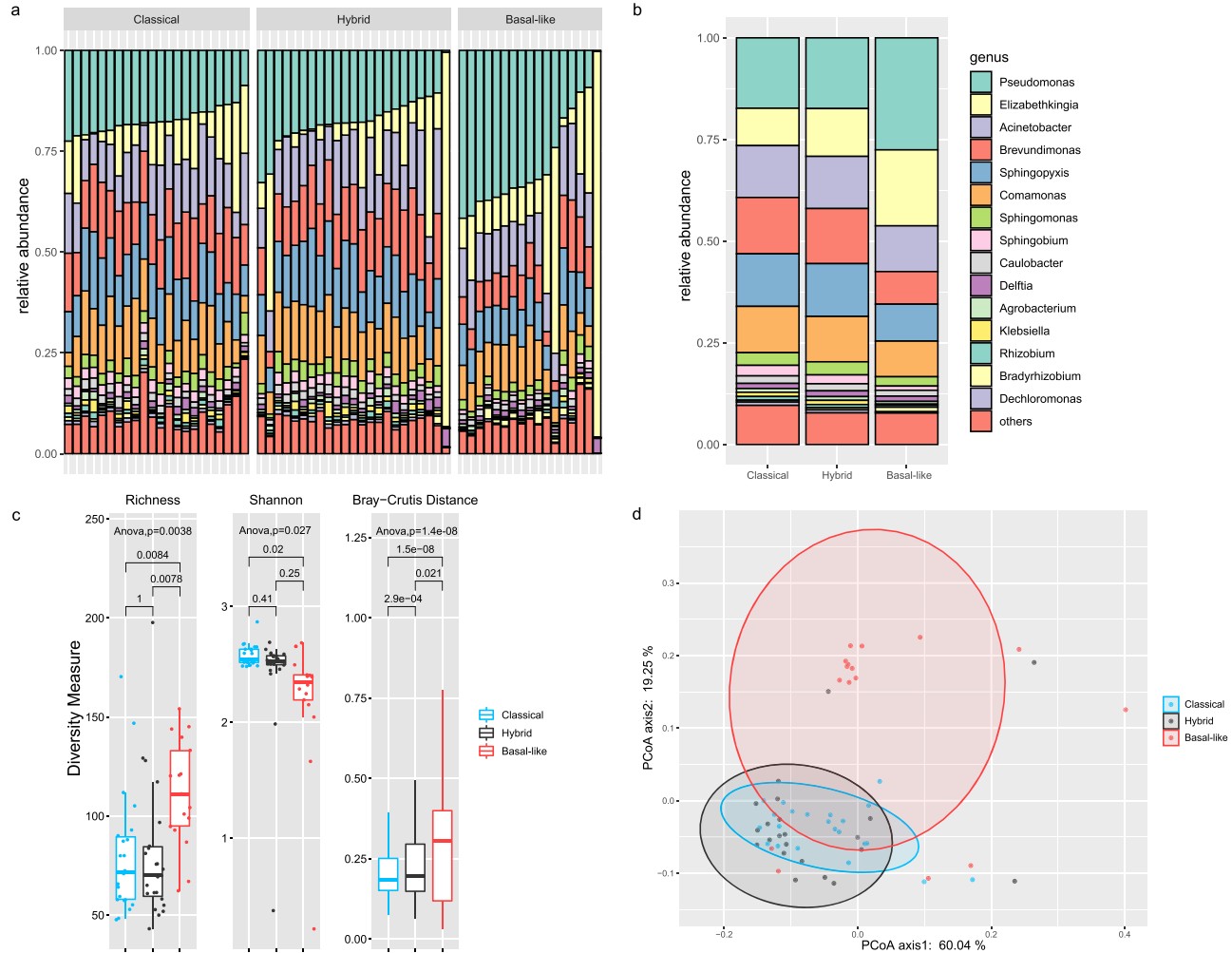

**Fig. 2 The tumor microbiome shows significant differences among PDAC subtypes. a–b** Barplots denote the relative abundance of predominant microbiota constituents at the genus level for each sample (**a**) and for each subtype (**b**). **c** Boxplots of microbial diversity measures at the genus level, including richness (observed genera), Shannon index and Bray-Curtis metric distance. *P* values were determined by ANOVA with post-hoc test. **d** PCoA plot of the tumor microbiome at the genus level based on Bray–Curtis dissimilarity. The ellipses indicate the 90% confidence interval.

identified 83 microbial genera that significantly differed in abundance across different tumor subtypes. Among these genera, most of them were significantly enriched in basal-like tumors, suggesting potential microbiome signatures for basal-like tumors. In contrast, microbial genera exhibiting enrichment in classical/hybrid subtype seemed to be highly abundant only in particular cases, indicating a negative finding of microbiome signature for these two subtypes (Fig. 3a, b). It was clearly demonstrated that basal-like tumors can be distinguished by a comparison heatmap based on the abundance of selected microbial genera. Among these basal-like enriched genera, three taxa with quite high abundances and extreme significances drew our attention. As shown in Fig. 3c–e, *Acinetobacter*, *Pseudomonas* and *Sphingopyxis* showed significant enrichment in basal-like tumors, suggesting that they may influence the tumor progression. We next stratified PDAC patients into high versus low groups based on the abundance of these three bacterial genera, and worse outcomes were predicted for PDAC patients with higher abundance of *Acinetobacter*, *Pseudomonas*, and *Sphingopyxis* (Fig. 3f–h).

We then attempted to expand these three bacterial genera to the species level and reveal the key species that may play crucial roles in PDAC progression. In this cohort, we identified four

bacterial species which show significant enrichment in basal-like tumors and belong to *Acinetobacter* genus: *Acinetobacter pittii*, *Acinetobacter junii*, *Acinetobacter baumannii* and *Acinetobacter haemolyticus*. In addition, seven bacterial species which belong to *Pseudomonas* genus (*Pseudomonas sihuiensis, Pseudomonas stutzeri, Pseudomonas alcaliphila, Pseudomonas pseudoalcaligenes, Pseudomonas sp. LPH1, Pseudomonas mendocina, Pseudomonas aeruginosa*) and five bacterial species which belong to *Sphingopyxis* genus (*Sphingopyxis macrogoltabida, Sphingopyxis fribergensis, Sphingopyxis granuli, Sphingopyxis sp. MG, Sphingopyxis alaskensis*) were identified to be enriched in basal-like subtypes (Supplementary Fig. 3). Significantly worse outcomes were predicted for PDAC patients with higher abundance of certain species, suggesting that the specific tumor microbiome has predictive value for PDAC prognosis (Supplementary Fig. 4).

To confirm the presence of intrapancreatic bacteria in PDAC cases, we next performed several additional experiments. A subset of Formallin-Fixed Paraffin-Embedded (FFPE) PDAC samples of our cohort were applied to the fluorescence in situ hybridization (FISH) using a specific probe that against bacterial 16S rRNA sequences. 16S rRNA FISH indicated the presence of bacterial DNA within PDAC tumors, and also demonstrated an increased

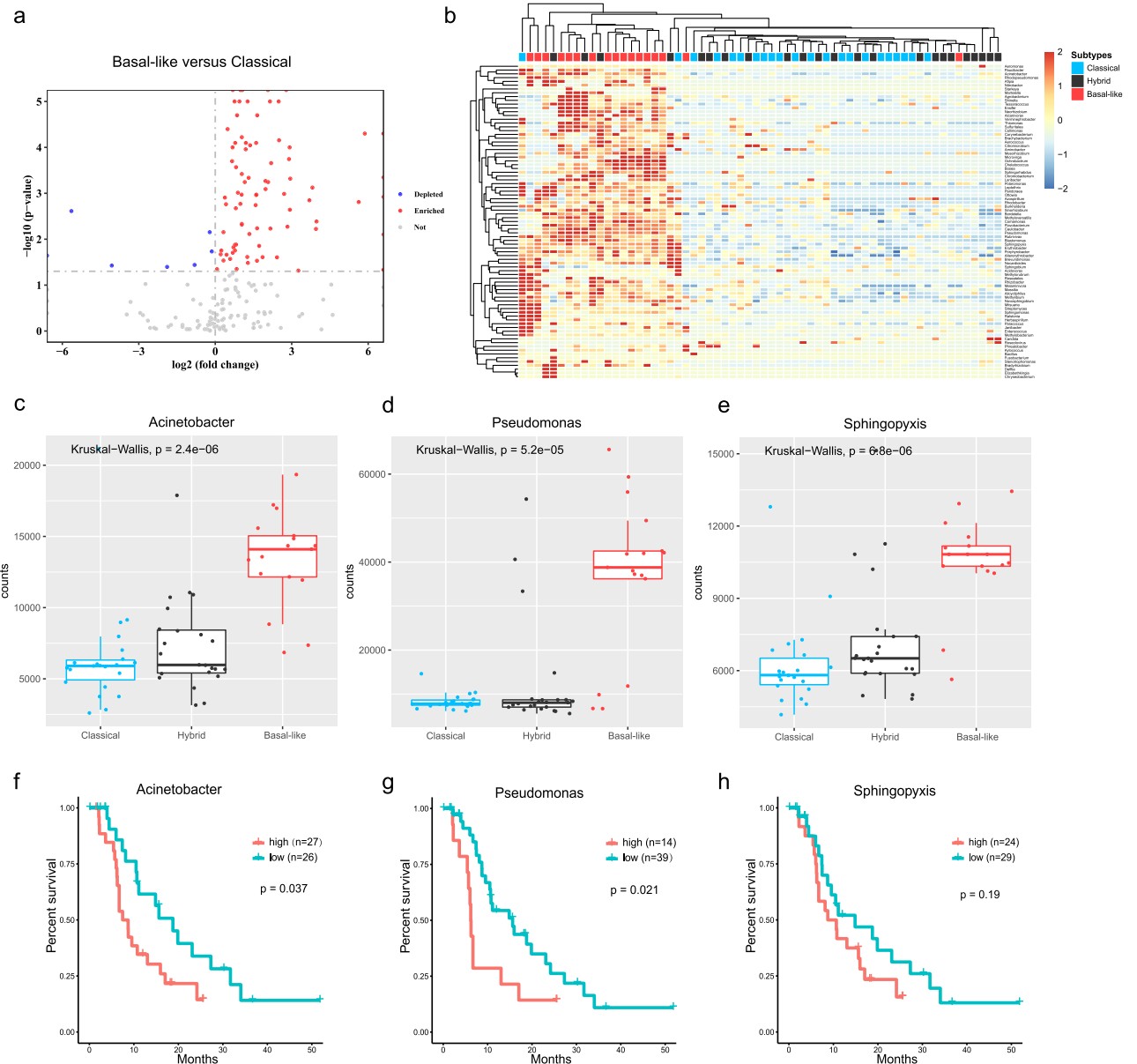

**Fig. 3 The distinctive microbial communities were characterized in basal-like tumors. a** Volcano plot showing differentially abundant microbial genera in basal-like versus classical tumors. A two-sided Wilcoxon rank-sum test was used to determine significance (*P* value < 0.05). **b** Heatmap of selected microbial genera with differential abundance across tumor subtypes; 83 genera (rows) are shown in the heatmap. **c–e** Abundance distribution of the genera *Acinetobacter* (**c**), *Pseudomonas* (**d**) and *Sphingopyxis* (**e**) in PDAC tumors. The *y*-axis indicates normalized counts. *P* values are from the Kruskal–Wallis test. **f–h**, Kaplan–Meier estimates for survival probability based on the abundance levels of three bacterial genera *Acinetobacter* (**f**), *Pseudomonas* (**g**), and *Sphingopyxis* (**h**). The log-rank test was used to determine significance.

load of bacteria in basal-like tumors compared with classical/ hybrid subtypes (Supplementary Fig. 5). Additionally, we conducted immunohistochemistry (IHC) using antibodies against bacterial lipopolysaccharide (LPS) to detect Gram-negative bacteria in PDAC tumors, as previously performed[19]. Consistently, bacterial LPS was detected in our FFPE PDAC samples, confirming the presence of intratumor bacteria in PDAC and displaying an increase in bacteria in basal-like PDAC (Supplementary Fig. 6). Furthermore, to validate the presence of tumor-related microorganisms and address the contaminations in our study, we conducted 16S rDNA PCR to detect the presence of target bacterial genera in PDAC tumors, pancreatic adjacent tissues and several controls. The data showed the corresponding amplification products of *Acinetobacter, Pseudomonas* and

*Sphingopyxis*, and the sequences turned out to match the 16S rDNA gene by Sanger sequencing (Supplementary Fig. 7). These results definitively confirmed the presence of *Acinetobacter, Pseudomonas* and *Sphingopyxis* in PDAC tumors, and largely proved that they were not laboratory-borne contaminations.

**The microbial communities enriched in basal-like tumors show inflammation-inducing potential.** Considering that numerous unknown microorganisms cannot be identified due to the incomplete database of microbiome genomes, we attempted to construct the microbial gene catalog in pancreatic tumors, which contributes to revealing the potential functional roles in the tumor microbiome of PDAC patients. We performed de novo

assembly and constructed metagenomic contigs based on non-human reads. Next, gene prediction was carried out, and a nonredundant microbial gene catalog that contains as many as 229,282 genes with an average length of 530 bp was established. We calculated the abundance of predicted genes using an in-house pipeline and identified differentially abundant genes through the Wilcoxon rank-sum test. Consistent with the previous findings in Fig. 3a, a large majority of differential genes were specifically enriched in basal-like tumors. A comparison heatmap based on 8960 basal-like enriched genes demonstrated a relatively distinct stratification between basal-like and classical/hybrid tumors (Fig. 4a). To further investigate the functional roles of microbial genes in tumor progression, functional characterization of genes was performed using KEGG ortholog (KO) and eggNOG ortholog groups (OG). The microbiome of basal-like tumors showed enrichment in many terms related to metabolism (transport and metabolism of amino acids, nucleotides, carbohydrates and lipids), energy production and conversion, replication, defense mechanisms and cell membrane/envelope biogenesis (Fig. 4b). At level 2 of the KEGG functional categories, increased levels of metabolism, cell motility, and drug resistance to antimicrobials were observed in basal-like enriched genes (Fig. 4c). These functional categories represent the microbial abilities of metabolic activity, cell motility and especially antibiotic resistance, reflecting an improved pathogenicity and hostile microbial environment. Our findings revealed the potential of the basal-like-related microbiome in inducing inflammation, which helps to elucidate the role of the specific microbiome in carcinogenesis.

**The microbiome in basal-like tumors is associated with carcinogenic gene expression programs.** The gut microbiota can influence intestinal physiology and disease by affecting tissue-specific transcription of host genes[27]. Similarly, we wondered whether aggressive progression and gene expression alterations are attributed to the specific microbiome in pancreatic cancer. For this purpose, we performed a general correlation analysis between microbial genera and host functional modules. Based on the differentially expressed genes identified in Fig. 1d, we constructed several gene modules exhibiting significant enrichment in certain terms of KEGG pathways, GO categories or hallmark gene sets. The association analysis demonstrated that some microbial genera, including *Acinetobacter*, *Pseudomonas* and *Sphingopyxis*, were positively correlated with functional modules of response to molecules of bacterial origin, response to lipopolysaccharide, and complement system, reflecting a natural antibacterial reaction by host immunity. In addition, we found that many cancer-associated functions, such as Kras signaling, EMT and MAPK signaling pathways also showed positive correlations with the aforementioned bacteria. In contrast, host functions related to bile acid metabolism, pancreatic beta cells and pancreatic secretion were found to be negatively correlated with the abundance of these genera (Fig. 5a). We presented a refined correlation network to summarily describe the relationship between 4 types of host functions and 3 key genera, which showed clear interplays that activated response to bacterial pathogens accompanied by increased levels of *Acinetobacter*, *Pseudomonas* and *Sphingopyxis*, were positively correlated with DNA replication and Kras signaling, and negatively correlated with bile acid metabolism (Pearson's correlation coefficient < −0.3 or >0.3, Fig. 5b). Furthermore, the correlation analysis between host functions and tumor microbiome was also conducted at the species level. Consistently, we found that the basal-like-enriched bacteria, especially the species belonging to *Acinetobacter*, *Pseudomonas* and *Sphingopyxis*, showed significant association with carcinogenic gene expression programs. The predominant bacterial

species *Acinetobacter junii*, *Pseudomonas sihuiensis*, and *Sphingopyxis fribergensis* displayed a positive correlation with Kras signaling, DNA replication and pancreatic cancer-related pathways (Supplementary Fig. 8). Together, these data revealed the influence of the tumor microbiome on host cell function and supported the crucial roles of the tumor microbiome in tumorigenesis.

**Correlation between host genetic variation and microbiome composition.** The composition of microorganisms in and on the human body varies widely across individuals and has been proven to be closely associated with host genetic variation[28,29]. The pivotal role of host genetic factors in shaping the gut microbial community has also been highlighted in recent years[30,31]. Based on these findings, we asked whether the diverse microbial compositions of PDAC tumors among patients were influenced by host genetic factors, which may confer a higher risk for the colonization of specific pathogenic microbiota and result in cancer progression. To this end, genome-wide genotyping of each individual was performed based on DNA sequencing data, and next, we estimated the genetic dissimilarity between individuals using Jaccard distance, a measure of how dissimilar two sets are, with a range from 0 to 1. We correlated microbial beta-diversity (Bray–Curtis metric distance) with genetic dissimilarity (Jaccard distance), and a significantly positive correlation was found ($R^2 = 0.14$, $p = 4.36e{-}07$, Fig. 6a). This result indicated that individuals with similar genetic variations were found to harbor similar microbial compositions, supporting that host genetics may play a role in shaping the microbiome.

To further explore the key genetic variations related to microbiome regulation, we performed a targeted analysis of the association between genetic variations and microbial genera. Several filtering steps were carried out to avoid the bias of testing (see methods). A total of 134 association pairs involving 97 SNVs and 29 genera were identified at $P < 5 \times 10^{-8}$ (Fig. 6b, Supplementary Data 3). The variation showing the strongest significance was located at chromosome 21 position 10,440,242, an intronic region of the BAGE gene, and was associated with the abundance of the genus *Paucibacter*. Notably, serveral SNVs related to gene CXCL1 were found a significant association with bacterial abundance. We next re-screened the SNVs that were associated with microbiome using a less stringent threshold ($P < 10^{-6}$), and 39 genes were identified from these loci. Next, we performed GO term and KEGG pathway enrichment analyses against the background of immune response-related genes. The functional enrichment for GO terms demonstrated that the potential microbial-regulatory genes are involved in interferon-γ-mediated signaling pathway and peptide antigen binding. Moreover, enrichment in KEGG pathways of Kaposi sarcoma-associated herpesvirus infection, natural killer cell-mediated cytotoxicity and TNF signaling pathways was also observed (Fig. 6c, d). To better visualize the interplay between host functional variation and the microbiome, we produced a correlation network consisting of 6 functional terms and 25 microbial genera (Fig. 6e). A similar targeted association analysis was also carried out at the species level, and 53 genes were identified from the QTLs associated with the abundance of species using the threshold of $P < 10^{-6}$. The functional enrichment analysis demonstrated a consistent result that these QTLs are involved in interferon-γ-mediated signaling pathway and infection-related pathways. The correlation network between host functional variation and the species was presented in Supplementary Fig. 9. Taken together, these findings indicated the potential association of immune-related functional alterations with microbiome compositions, which generated a possible

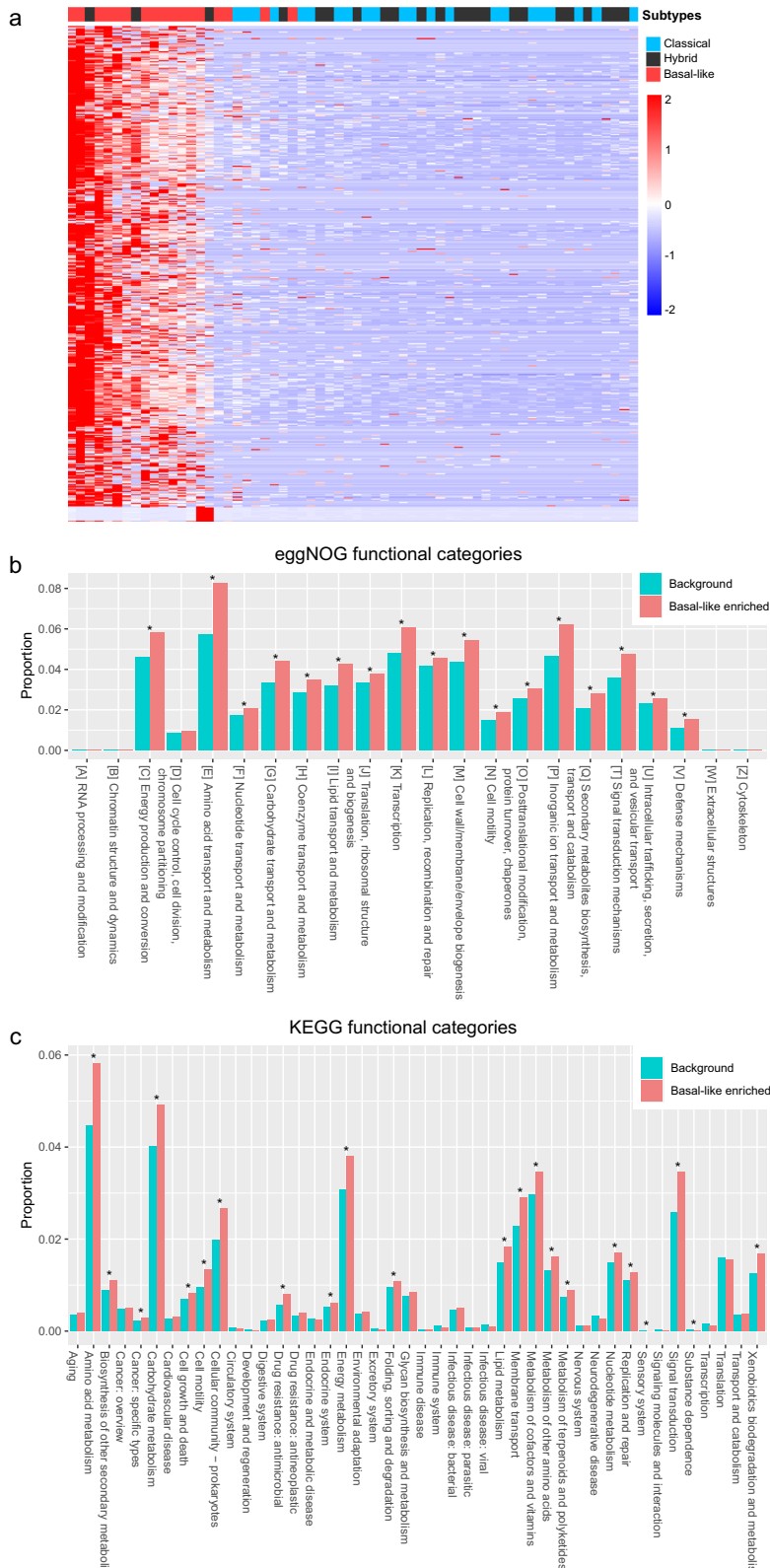

**Fig. 4 Microbial genes enriched in basal-like tumors display higher pathogenicity. a** Heatmap of microbial genes with differential abundance across all PDAC samples. The heatmap contains 8960 genes (rows) and 62 samples (columns). **b** Comparison between the Basal-like enriched and background genes on OG functional categories. **c** Comparison between the Basal-like enriched and background genes on level 2 of KEGG functional categories. Significance was determined by Fisher's exact test.

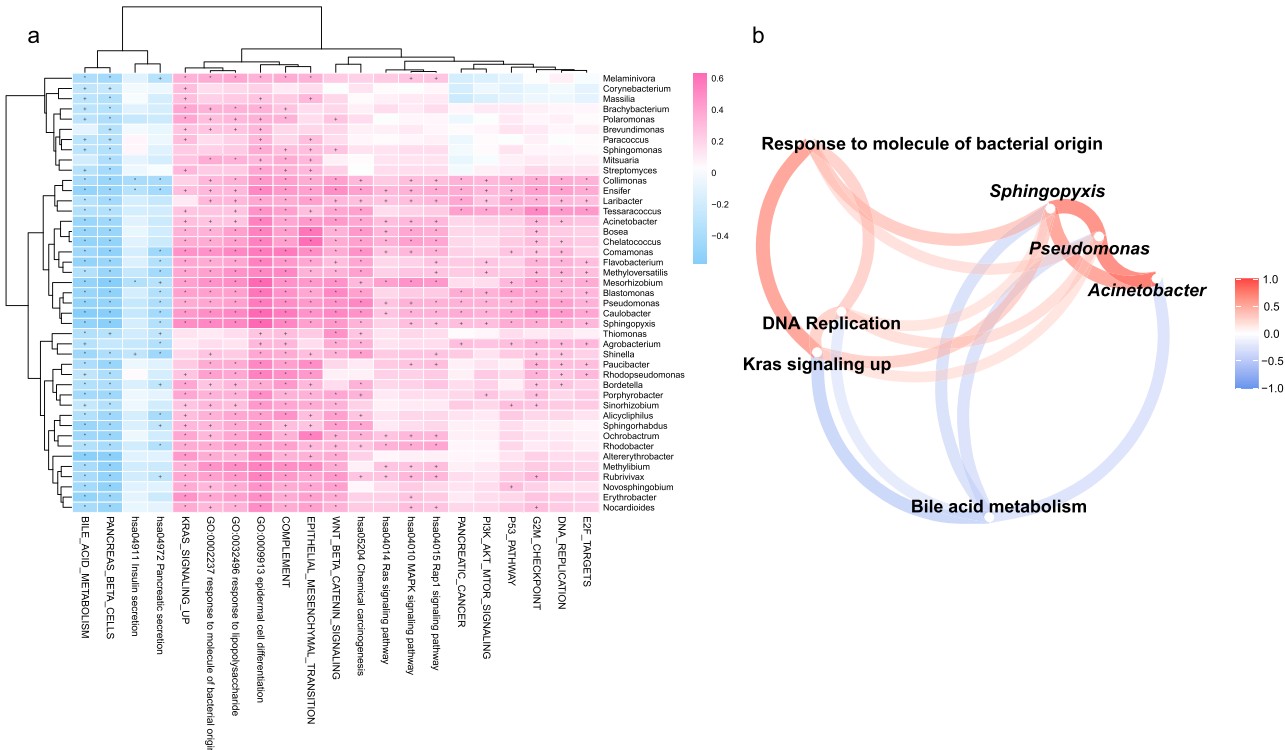

**Fig. 5 The tumor microbiome is associated with host functional modules. a** Heatmap of Pearson's correlation between 21 functional modules (columns) and 43 bacterial genera (rows). The color represents the coefficient of correlation. + denotes the $p$ values <0.05, and * denotes the $p$ values <0.01. **b** Correlation network between 4 functional modules and 3 key genera refined from (**a**). Red and blue edges denote Pearson's correlation coefficients >0.3 and <−0.3, respectively.

relationship between genetic variations and microbiome communities.

## Discussion

Cancer is classically considered a disorder attributed to aberrations in the genome[32]. However, emerging evidence has shown that the microbiome also makes substantial contributions to some types of cancer; for example, the gut microbiota was demonstrated to play a regulatory role in liver cancer via bile acid metabolism[33]. In other sites beyond the gut, microbial dysbiosis is being increasingly recognized for its role in oncogenesis[34]. Our study presents the first research to explore the influence of the tumor microbiome in different PDAC subtypes. Current subtyping schemes can identify prognostic subgroups by transcriptome analysis[23]. Guided by these studies, our samples were split into three subtypes labeled 'basal-like', 'classical', and 'hybrid'. Basal-like tumors are thought to be more aggressive, with worse outcomes[6]. The main distinguishing genes (signature 2) are related to squamous differentiation, suggesting that basal-like tumors may involve the squamous malignancies. Squamous expression programs usually accompany TP53 and KDM6A mutations and exhibit poor prognosis[35]. Comparatively, the default path of pancreatic pathogenesis is considered to cause the classical phenotype. The signatures for classical tumors are related to pancreatic lineage differentiation. Clinically, classical tumors are more frequent in early stage[23]. The single-cell analysis demonstrated that most tumors harbor both basal-like and classical programs, which lead to a mixed transcriptional profile at the bulk RNA-seq level, and hybrid tumors may be the outcome of this issue[23]. Consistent with these findings, we found that basal-like tumors were enriched for DNA replication, TGF-β signaling, and EMT. Interestingly, our data demonstrated that the

inflammatory response and interferon-γ response were elevated in basal-like tumors. The immune infiltration analysis also showed an increasing level of activated mast cells and M1 macrophages in basal-like tumors. M1 macrophages are able to secrete pro-inflammatory cytokines and play roles in direct host-defense against pathogens[36]. Additionally, mast cells interact directly with bacteria and are involved in allergic inflammation[37]. All these signs indicate that pathogen-induced inflammation is critical for aggressive progression in basal-like tumors.

Inflammation is relevant as a risk factor for PDAC development and progression. Lipopolysaccharide (LPS) can hyperstimulate Kras and lead to the initiation of carcinogenesis[38,39]. We inferred that the tumor microbiome may produce procarcinogenic effects through perpetual inflammation induction rather than direct mutagenic effects. In this study, we identified distinctive microbial communities in basal-like tumors, and notably increasing levels of *Acinetobacter*, *Pseudomonas* and *Sphingopyxis* were observed. *Acinetobacter* and *Pseudomonas* are common opportunistic pathogenic bacteria and are able to cause serious infections[40,41]. Notably, the increased abundance of *Pseudomonas* was also observed in PDAC patients with short-term survival by Riquelme et al. study, which confirmed the protumorigenic effect of *Pseudomonas* as revealed in our study[18]. Besides, Pushalkar et al. identified *Pseudomonas* as the most abundant bacterial genus that translocated from gut to pancreas in PDAC patients[17]. As demonstrated by the comparison of pancreatic microbiome between PDAC and wild-type mice, higher level of *Acinetobacter* was observed in PDAC mice. The presences of predominant genera such as *Elizabethkingia*, *Delftia*, *Agrobacterium*, and *Sphingomonas* by our analysis were also detected in PDAC patients by previous studies[17,18]. The microbial family *Sphingomonadaceae* (the upper level of *Sphingopyxis*) is a gram-negative bacterium that contains glycosylceramides in the outer

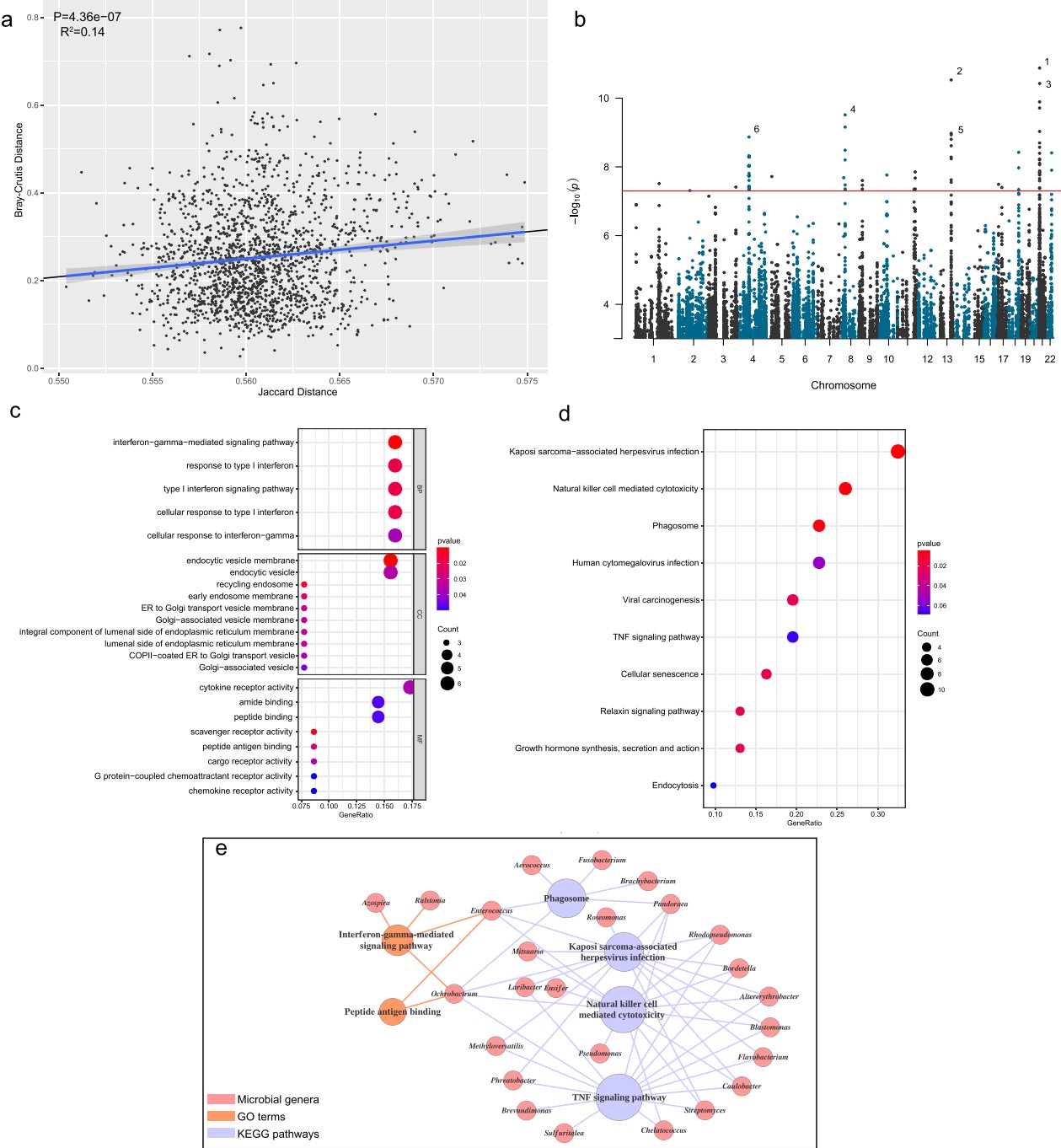

**Fig. 6 Host genetic variation can influence the composition of the tumor microbiome. a** Genetic dissimilarity defined by Jaccard distance between patients is correlated with microbial beta-diversity (Bray–Curtis metric distance). Linear regression was used to evaluate the significance and coefficient of determination. The 95% confidence interval is shown. **b** Manhattan plot illustrating QTLs for microbial genus abundance. Each dot denotes an association pair between genetic variation and a microbial genus. The number labeled beside denotes the significance rank of QTLs. The red line corresponds to a significance threshold of $5 \times 10^{-8}$ (FDR < 0.01) **c** Gene ontology enrichment for genes identified from loci associated with the abundance of genera. **d** KEGG pathway enrichment for genes identified from loci associated with the abundance of genera. The color and size of each bubble denote enrichment significance and the number of genes enriched in the functional category, respectively. **e** Correlation network showing the association between selected enriched GO terms or KEGG pathways and microbial genera. The functional terms were manually classified into 3 types. Lines denote the association, and colors represent the types of functions.

membrane and can be detected in some human fecal samples but is not abundant or highly pathogenic[42]. NKT cells were shown to be stimulated after exposure to bacterial glycosphingolipids from *Sphingomonadaceae*, suggesting a classical innate immune response by pattern antigen recognition[43,44]. In addition, we performed functional characterization of the microbial genes

enriched in basal-like tumors and found enrichment of metabolism, energy production, replication, cell motility, defense mechanisms and cell membrane/envelope biogenesis, indicating improved microbial activity and pathogenicity[45]. These findings revealed the inflammation-inducing role of the specific microbiome in basal-like tumors. Importantly, we found few

enrichments of microbiata in classical/hybrid subtype, which may imply that abundant presence of microbiota in pancreas likely generates a protumorigenic effect rather than a common commensal role.

It has been reported that microbiota dysbiosis can induce transcriptional alterations in intestinal epithelial cells[27]. We next aimed to explore the relationship between the host and tumor microbiome. Our data demonstrated that the abundance of specific bacteria, including *Acinetobacter*, *Pseudomonas* and *Sphingopyxis*, positively correlated with host functions of the antibacterial immune response (complement system, response to lipopolysaccharide, and response to molecules of bacterial origin), presenting the interaction between microbial invasion and host immunity. Moreover, many cancer-associated signaling pathways, such as Kras signaling, MAPK signaling and the Wnt/β-Catenin pathway, also displayed significant correlations with these microbiota constituents. The facilitation effect of inflammatory insults, such as LPS in Kras signaling, has been described previously[38,39]. Experimental evidence also suggested that bacterial infection can induce malignant transformation by activating the AKT and MAPK signaling pathways[46]. Interestingly, our data demonstrated that the pancreatic tumor microbiome may give rise to dysfunction of bile acid metabolism. The role of bile acid in liver cancer has been widely studied, and the gut microbiota has an important impact on the composition of bile acids[33]. However, the role of bile acid in pancreatic cancer remains unclear. Our work provided some insights into the effect of the pancreatic microbiota on bile acid metabolism and pancreatic cancer. Collectively, this study depicted a detailed relationship between host cell function and the tumor microbiome, which helps to elucidate the tumorigenic role of the specific microbiota.

An essential question is what determines the microbiome composition. We wondered whether host genetics have impacts on shaping diverse microbial communities, which may confer a higher risk for the colonization of the specific pathogenic microbiota and result in carcinogenesis. Our findings demonstrated the remarkable relevance of host genetic factors and microbiome compositions, proposing a possible role of host genetics in shaping the microbiome. In addition, we found that genetic variations related to immune dysfunction, such as compromised function of interferon-γ signaling or NK cell-mediated cytotoxicity, are associated with the abundance of bacteria. This analysis led to a possible hypothesis that independent of microenvironment conditions, host antimicrobial immunodeficiency may induce microbiome imbalances and give rise to the invasion of a specific pathogenic microbiota that promotes carcinogenesis.

It is important to note that our cohort is comprised only of resectable pancreatic tumors due to difficulty in obtaining specimens from metastatic patients. This caused a limitation that our finding may be not applicable to advanced stage PDAC with metastatic tumor. Further study is required to take entire cancer population into the consideration. Interestingly, the tumor microbiome seems to have the dual effect of promoting immune suppression or activation by previous studies. Pushalkar et al. reported that ablation of the tumor microbiome was protective against cancer progression via immune activation[17]. However, several studies suggested that the presence of microbiome in gut/tumor may boost the cancer immunity and be linked to favorable immunotherapy responses or tumor-fighting efforts[18,26]. The complex interplay among microbiome, immune cells and pancreatic tumorigenesis remains unsolved. It is quite difficult to answer the precise mechanisms of microbiome in immunity by our results. In this study, we focused on the effect of persistent inflammation induced by tumor microbiome in PDAC, and suggested that the activated inflammatory response in basal-like subtype serves as a facilitator in pancreatic carcinogenesis. In conclusion, our study delineated the intrapancreas microbiome profile of resected PDAC and found that the distinctive microbial communities in basal-like tumors may play an inflammation-inducing role and contribute to pancreatic carcinogenesis. We detailed the interplay between host cell function and the tumor microbiome, suggesting the tumorigenic role of specific microbiome compositions. We also demonstrated the effect of host genetics on PDAC microbiome. Our work highlighted the microbial basis of PDAC heterogeneity and supported the potential that microbial signatures may be useful as predictors of patient outcomes. Treatment targeting the PDAC microbiome may be promising.

## Methods

**Patients and samples.** Patients with a confirmed PDAC diagnosis were recruited from ChangHai Hospital, Shanghai. All patients gave written informed consent for collection and use of the samples. In this study, patients who received antibiotic treatment within the past month were excluded. A total of 62 resectable tumors were obtained from surgical specimens and immediately frozen at −80 °C prior to nucleic acid extraction. All samples were collected between 2015 and 2016, and only fresh tumors were used in this study. All procedures conformed to the Code of Ethics of the World Medical Association (Declaration of Helsinki) and were complied with the guidelines of the institutional review committee of the Shanghai Institute of Nutrition and Health, Chinese Academy of Sciences.

**DNA/RNA extraction and sequencing.** Frozen tumor tissues were aseptically sliced into appropriate sizes and placed in 1.5 ml sterile tubes. We utilized β-mercaptoethanol and a rotor-stator homogenizer to disrupt the tissue and homogenize the lysate. Metagenomic DNA and total RNA were simultaneously extracted using QIAGEN Pathogen Lysis Tubes and QIAGEN AllPrep DNA/RNA Mini Kit. DNA/RNA was quantified for concentration and purity using a Nano-Drop spectrophotometer and stored at −20 °C. DNA samples were applied to library construction and whole-genome sequencing on the Illumina platform and yielded 2 × 150 bp paired-end reads. After discarding the adaptor contamination and low-quality reads, an average of 80 Gb of clean data per sample was obtained. Meanwhile, samples of RNA were sequenced for the whole transcriptome on the Illumina platform (paired-end, 2 × 150 bp reads) and yielded an average of 15 Gb of clean data per sample.

**RNA-seq analysis.** Clean RNA sequencing data from 62 pancreatic tumors were aligned into human genome reference (GRCh38) using HISAT2 with default options[47], and then we used HTseq to count reads for each gene based on the GTF file, which provides gene structure information[48]. The gene count matrix was normalized using DESeq2 size factors determined by median ratios[49]. Differential gene expression analysis between different subtypes (basal-like versus classical) was performed with DESeq2 based on the negative binomial distribution model. The significantly differentially expressed genes ($p$ value <0.01) were ranked based on the log$_2$ fold change. Gene set enrichment analysis (GSEA) was next performed against the hallmark gene set from MSigDB using the R package clusterProfiler[50,51]. The normalized enrichment score was assessed for each set.

**Tumor subtype clustering.** Guided by the Michelle et al. study[23], we performed consensus clustering to identify tumor subtypes using the tumor expression signatures from their non-negative matrix factorization (NMF) results. We selected the top 25 ranked gene signatures from each of the four malignancy-related NMF components (1, 2, 6, and 10) and utilized the R package ConsensusClusterPlus for tumor subtyping[52]. For clustering, we used Pearson correlation distance and the hierarchical cluster algorithm with 10,000 resamplings to generate the final consensus. According to the cumulative distribution function (CDF) plot, a flatter middle segment represented a lower value of the proportion of ambiguous clustering (PAC). Therefore, we chose $k = 3$ as the optimal clustering number. Finally, three main subtypes were identified in our cohort.

To validate the clustering result, we performed another subtyping using gene signatures from the Moffitt et al. study[6]. The top 25 ranked genes from each of two NMF components (basal-like and classical) were selected. Consensus clustering was performed with the same parameters as in the previous method, and the samples were divided into 3 different subtypes based on the optimal CDF.

**Evaluation of immune cell infiltration by CIBERSORTx.** CIBERSORTx is a computational approach that accurately resolves the relative fractions of diverse cell subsets using gene expression data[53]. Combined with the leukocyte gene signature matrix, which is termed LM22, CIBERSORTx was used to estimate the fractions of 22 immune cell types in our study. The differential expression proportion of immune cell types was evaluated using the Kruskal-Wallis test across tumor subtypes.

**Survival analysis**. Overall survival was defined as the time from tumor resection to death from any cause. Patients who did not experience death were censored at date of last follow-up. The survival information of PDAC patients were collated, and some clinical data were obtained (14 basal-like, 21 hybrid, and 18 classical). We utilized the log-rank test to assess the difference in survival between different groups.

**Bacterial 16S rDNA PCR, 16S rRNA FISH and LPS immunostaining assays**. For 16S rDNA PCR, metagenomic DNA from PDAC tumors was detected using genus-specific primers targeting *Acinetobacter*, *Pseudomonas* and *Sphingopyxis*. The primer sequences used in this study were: 5'-CGGACGGGTGAGTAATGCTT-3', 5'-CAGACCCGCTACAGATCGTC-3' (*Acinetobacter*); 5'-CAAGGCGACGATCC GTAACT-3', 5'-ATGGCTGGATCAGGCTTTCG-3' (*Pseudomonas*); 5'-AGGCGA CGATCTTTAGCTGG-3', 5'-ACGCCCAGTAATTCCGAACA-3' (*Sphingopyxis*). We introduced four types of negative controls to address contaminations in this study, including DNA from pancreatic adjacent tissues using the same extraction process, environmental control (samples from the same freezer), DNA extraction control (buffers of extraction kit), and PCR no-template control.

Ribosomal RNA (rRNA) fluorescence in situ hybridization (FISH) was executed using the EUB338 16S rRNA gene probe labeled with the fluorophore 5'-Cy3 (extinction wavelength, 555 nm; emission wavelength, 570 nm; Molecular Probes)[54]. FFPE tumor tissues were hybridized to detect the bacterial colonization within pancreatic tissues.

FFPE PDAC tissues were stained for bacterial LPS (Lipopolysaccharide Core, mAb WN1 222-5, HycultBiotech, 1:1000 dilution) with the automated slide stainer BOND RXm (Leica) using the Bond polymer refine detection kit. Heat induced epitope retrieval was done by a 20 min heating step with the epitope retrieval solution 1 (BOND).

**Microbial taxonomic assignment and abundance calculation**. After sequence trimming and duplicate filtering by Fastp[55] and Super-deduper (https://github.com/dstreett/Super-Deduper), the passing reads were aligned to the human reference (GRCh38) using Bowtie2[56] to preliminarily remove the host DNA sequences. The remaining reads were processed for taxonomic assignment using Kraken2[57]. A customized database, which consists of reference libraries of bacteria, viruses, fungi, archaea, plasmids, UniVec and human from the NCBI database, was constructed for taxonomic classification in Kraken2. To avoid mistaken matching generated by repeated sequences, a confidence score threshold of 10% was set to improve the accuracy of classification. Combined with the Kraken2 classifier, we utilized Bracken[58] to estimate the counts of reads originating from every taxon present in the sample based on Bayesian probability and produced a relatively accurate abundance profile at the species, genus, family, order and class levels. The microbial count matrix was normalized by DESeq2 sizefactors[49], and the relative abundance of the microbiome was calculated. Alpha-diversity (Shannon index) and beta-diversity (Bray–Curtis metric distance) analysis and principal coordinates analysis (PCoA) utilized the R package vegan[59]. Differential abundance analysis of the microbiome utilized the Kruskal-Wallis test among tumor subtypes. The false discovery rate (FDR) was used to correct for multiple hypothesis testing.

**Microbial gene catalog construction and functional characterization**. Reads without human-associated contaminants were assembled by the MEGAHIT assembler, which shows improved performance for large and complex metagenomics assembly[60]. We utilized MetaGene[61] to predict prokaryotic genes in contigs with lengths larger than 300 bp. A nonredundant gene catalog was constructed by removing redundancies that aligned to others with over 95% identity and more than 90% coverage using Cd-hit[62]. Taxonomic assignment of predicted genes was performed using Diamond[63] with the BLASTX algorithm against the NR database of NCBI. Alignment hits with e-values larger than 1e−5 or query coverage less than 80% were filtered, and the best match was retained for the subsequent annotation of each gene. If one gene was equally matched to multiple alignments with the same score, the taxonomic assignment was determined by the lowest common ancestor (LCA) algorithm. The abundance of the predicted gene was estimated using an in-house pipeline in which nonhuman reads were aligned against the customized nonredundant gene catalog using Bowtie2, and the gene length was used to calculate the abundance by dividing counts. Differential abundance analysis of predicted microbial genes was performed using a two-sided Wilcoxon rank-sum test, and p values were adjusted with the FDR algorithm. The functional characterization of predicted genes was performed on the basis of KEGG ortholog and eggNOG ortholog groups with the application of BlastKOALA and eggNOG-mapper, respectively[64,65]. Due to the very low number of basal-like depleted genes, we compared the gene proportion of each functional category between the basal-like enriched gene set ($n = 8960$) and background gene set (all microbial genes, $n = 229,282$). Significance was determined by Fisher's exact test.

**Correlation analysis between host functional modules and the microbiome**. On the basis of differentially expressed genes identified in RNA-seq analysis, we performed gene enrichment analysis against several functional categories, including KEGG pathways, GO terms and the hallmark gene set, using a hypergeometric test by clusterProfiler[51]. The level of functional modules was identified in three steps. First, the significant enrichment categories were selected according to the $p$ values ($p < 0.05$). Second, for each enrichment category, the enriched genes were used to build the expression profile. Third, we estimated the first principal component (PC1) to represent the general expression level of the functional module. Microbial genera with an abundance of zero in more than 80% of samples were excluded, and 122 genera were left for the correlation analysis. The correlation coefficient was calculated using the Pearson algorithm in the R package psych, and $p$ values were adjusted by FDR for multiple comparisons. Visualization was performed using the R package corrr.

**Genotype calling, filtering and annotation**. Genetic profiles in the form of single nucleotide variants (SNVs) and short insertion/deletions (InDels) were identified using the Genome Analysis Toolkit (GATK) version 4.0 pipeline. Clean DNA sequencing data from 62 pancreatic tumors were aligned to human reference (GRCh38) using BWA-MEM with default parameters[66], and we utilized Picard tools to process the postalignment procedures, including sorting and indexing (https://broadinstitute.github.io/picard/). Briefly, the bam files yielded by alignment were submitted to mark duplicates, base quality score recalibration (BQSR), SNV calling, and variant filtration by the Java programs in GATK. Over 3,500,000 SNVs per sample were identified after quality control, and we utilized ANNOVAR[67] to perform gene annotation based on the ENSEMBL database.

**Association analysis between host genetics and the microbiome**. First, a large integrated genotype matrix of all 62 PDAC samples was constructed based on the SNV profiles, which consisted of 14,692,240 rows (SNVs) and 62 columns (samples). We excluded SNVs with minor allele frequencies (MAFs) less than 0.1, resulting in a smaller genotype profile with 4,778,023 SNVs. Jaccard distance, also known as the complement of the intersection over union, was calculated based on the genotype matrix in R using vegan[59]. We used linear regression to evaluate the correlation between Jaccard distance and microbial Bray–Curtis metric distance in R.

To link the microbial genera to genetic variation, we treated the abundance of genera as quantitative traits, and quantitative trait locus (QTL) mapping was carried out using the R package MatrixEQTL[68], which provides ultra-fast eQTL analysis for large matrix operations. We chose a linear model based on the assumption that genotypes have only additive effects on microbial abundance. To reduce the multiple test burden, we defined a region of 250 kb around the immune-related genes according to the gene list (total 770 genes) supplied by NanoString[69]. We selected SNVs located at this region for the targeted association analysis. After filtering, a final genotype matrix consisting of 171,561 targeted SNVs was yielded for downstream analysis. Only microbial genera present in at least 20% of samples were included. For the association analysis, about $10^7$ pairwise tests were performed between roughly $10^5$ SNVs and approximately 100 microbial genera. We identified significant associations using a stringent threshold of $P < 5 \times 10^{-8}$, which corresponds to the FDR < 1%. We found 134 associations at this level of significance.

In order to investigate the impact of identified SNVs on function. We re-screened the associations with a less stringent cutoff of $P < 10^{-6}$, and a total of 340 associations were found. The SNVs were annotated to genes based on their chromosome locations, and a total of 39 potential microbial-regulatory genes were identified. We performed GO term and KEGG pathway enrichment analyses against the particular background gene list from NanoString[69], which was used for screening in the previous steps. We linked the functional category to microbial genera if the enriched genes from this functional category were associated with the microbial genera. The correlation network was visualized by Cytoscape version 3.6[70].

**Statistics and reproducibility**. All statistical analyses were carried out using R v3.6.3 with corresponding packages. $P$ values < 0.05 were considered significant. The FDR was used to adjust p values for multiple hypothesis testing. The selections of statistical methods were described in each steps.

**Reporting summary**. Further information on research design is available in the Nature Research Reporting Summary linked to this article.

## Data availability

All sequence data of RNA-seq have been uploaded to NCBI Gene Expression Omnibus database under accession number GSE172356. The raw data for microbiome analysis is available on NCBI BioProject Accession Number: PRJNA719915. The raw unedited gel images are provided as Supplementary Figures. The processed data are available in Supplementary Data 1–3. Source Data are provided in Supplementary Data 4.

## Code availability

The custom codes used in this study are provided via Github at https://github.com/withered-leaf/TM-pipeline.

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

## Acknowledgements

This work was supported by the National Key Research and Development Program of China (2017YFA0103501), Special Fund for Strategic Pilot Technology of Chinese Academy of Sciences (XDA12010204), Key Program of the Chinese Academy of Sciences (QYZDJ-SSW-SM01), General Program of National Natural Science Foundation of China (81570827, 31471224), Innovation Program of Shanghai Municipal Education Commission (2017-01-07-00-07-E00012), and Shanghai Science and Technology Committee (20511101200).

## Author contributions

W.G.: Conceptualization, analysis, investigation, methodology, writing-original draft. Y.C.Z.: investigation, methodology, software. S.G.: conceptualization, resources. Z.M.: investigation, validation. H.L.: investigation, validation. H.D.: data curation, investigation. K.W.: investigation. H.Y.: investigation. Y.H.Z.: investigation. Y.Z.: investigation. J.L.: investigation. L.H.: conceptualization. J.G.: resources. X.K.: conceptualization, supervision, project administration, writing-review, and editing.

## Competing interests

The authors declare no competing interests.
