## [Transparent Peer Review File · Communications Biology]

Reviewers' comments:

Reviewer #1 (Remarks to the Author):

The manuscript by Guo et al. addresses a very interesting field of microbiome alterations in different molecular subtypes of PDAC. The authors show a distinct microbiome signature in basal-like tumors that is characterized by an increased abundance of certain bacteria such as *Acinetobacter*, *Pseudomonas* and *Sphingopyxis* potentially associated with increased inflammation and carcinogenic potential as suggested by several pathway analyses.

Although the findings are compelling and the manuscript is nicely written, there are several major limitations of the study that impair the overall quality of the study and do not allow meaningful conclusions at present:

1) The authors present a relatively small cohort of resected PDAC patients (n=68) which are then divided in 3 subgroups (basal, classical and hybrid). The hybrid subtype is not an established molecular subtype with known biological and clinical consequences. Therefore, the hybrid subtype appears arbitrarily determined and not valid. Furthermore, the poor prognosis of the basal like subtype should be shown by providing clinical data (progression free survival, overall survival etc) of the cohort. As surgical samples were taken 2015-2016, the majority of patients data should be available.

2) A major concern of the conducted microbiome experiments is the complete lack of any negative controls. Neither non tumoral-tissue (e.g. adjacent duodenum) nor elution or library preparation buffers are used to account for likely contaminations that occur in almost any of these microbiome experiments. To this end, a recent paper published by Nejman et al. (PMID: 32467386) sets a need standard regarding the required controls. The reported significant associations could otherwise be purely coincident findings without meaningful biological implications

3) The displayed genus in Figure 2b are predominantly present in the environment that occur in water, paper mill waste (e.g. *Dechloromonas*) and freshwater in Korea (*Sphingopyxis*). *Sphingopyxis* has never been shown to be present in human organisms. This again requires negative controls from samples stored in the same freezer to avoid likely cross-contamination. The absolute requirement of including negative controls in any microbiome analysis is nicely summarized in Rob Knight et al. (Nature Reviews Microbiology 2018).

4) The real presence of the top bacteria in basal like tumors should be confirmed by independent methods such as PCR, FISH or IHC.

5) As metagenomics sequencing is conducted, the author should attempt to analyze species not genera. The explanation in lines 169-171 is therefore not sufficient.

Minor:

1) Line 50: wrong citation, no 13 fold increased risk is reported in this paper

2) Line 67: not Erick et al. but Riquelme et al (see citation 18)

3) Figure 2c: wrong statistical test: Anova + post-hoc tukey should be used here (given that normal distribution and homogeneity of variance is present)

4) Figure 3c-e: y axis: which count is meant: read count? This depends of factors such as DNA input, Please show relative abundance

Reviewer #2 (Remarks to the Author):

This is a well written manuscript that would be a welcomed and major contribution linking the fields of tumor genomic heterogeneity with microbiome.

The data is very nicely presented and the text is clear.

My only criticism is that the authors need to temper their language implying a cause and effect relationship between the microbial and genomic findings or vice versa. There is not enough information to imply causality. Regardless, this is a major paper that will be highly cited. The human dataset is impressive and builds on an important body of literature.

Reviewer #3 (Remarks to the Author):

In "Tumor microbiome contributes to an aggressive phenotype in the basal-like subtype of pancreatic cancer" Guo et al associate distinct subtypes of pancreatic cancer with their unique intratumoral microbial signatures. The manuscript is well written, premise is solid and investigates important interactions between tumor microbiota and tumor biology. I have the following questions:

1. The "mycobiome" has recently been shown to modulate pancreatic cancer (Aykut et al, 2019). Did the authors check for any fungus in the tumors?
2. How did the authors control for laboratory contamination while looking for microbes inside the tumors? Did they use any negative controls for example, adjacent normal tissues?
3. What were the baseline characteristics of subjects? Did they have a recent history of antibiotic intake or instrumentation as these can change the intrapancreatic microbial burden within various subtypes?
4. Many dead bacterial products may be carried to the pancreas by the phagocytes and sequencing results may not be able to differentiate dead microbial DNA from that of live microbes. Did the authors try to culture pancreatic tumors to see if basal-like subtype indeed has increased growth compared to other subtypes?
5. Please mention other important works dealing with pancreatic cancer and the microbiome (Aykut et al, 2019; Sethi et al, 2018; Nejman et al, 2020). How did the author's findings compare to the tumor microbiota presented in Nejman et al and Riquelme et al ?

Reviewers' comments:

Reviewer #1 (Remarks to the Author):

The manuscript by Guo et al. addresses a very interesting field of microbiome alterations in different molecular subtypes of PDAC. The authors show a distinct microbiome signature in basal-like tumors that is characterized by an increased abundance of certain bacteria such as *Acinetobacter*, *Pseudomonas* and *Sphingopyxis* potentially associated with increased inflammation and carcinogenic potential as suggested by several pathway analyses.

Although the findings are compelling and the manuscript is nicely written, there are several major limitations of the study that impair the overall quality of the study and do not allow meaningful conclusions at present:

Reply to Reviewer #1

Dear Reviewers,

Thank you very much for your constructive suggestions to improve the quality of our manuscript. We appreciate your detailed feedback and have supplemented several data to make our results convincing according to your comments. The detailed point-by-point responses are listed below.

1) The authors present a relatively small cohort of resected PDAC patients (n=68) which are then divided in 3 subgroups (basal, classical and hybrid). The hybrid subtype is not an established molecular subtype with known biological and clinical consequences. Therefore, the hybrid subtype appears arbitrarily determined and not valid. Furthermore, the poor prognosis of the basal like subtype should be shown by providing clinical data (progression free survival, overall survival etc) of the cohort. As surgical samples were taken 2015-2016, the majority of patients data should be available.

Response: According to the comment, we have added contents to detailedly describe the definition of different subtypes in Discussion. It should be noted that the hybrid subtype of PDAC has also been mentioned in previous publication. A plausible explanation was proposed for the unclear signatures of hybrid subtype.

We added in Discussion in page 14, line 340-350: "*Basal-like tumors are thought to be more aggressive, with worse outcomes [6]. The main distinguishing genes (signature 2) are related to squamous differentiation, suggesting that basal-like tumors may involve the squamous malignancies. Squamous expression programs usually accompany TP53 and KDM6A mutations and exhibit poor prognosis [35]. Comparatively, the default path of pancreatic pathogenesis is considered to cause the classical phenotype. The signatures for classical tumors are related to pancreatic lineage differentiation. Clinically, classical tumors are more frequent in early stage [23]. The single-cell analysis demonstrated that most tumors harbor both basal-like and classical programs, which lead to a mixed transcriptional profile at the bulk RNA-seq level, and hybrid tumors may be the outcome of this issue [23].*"

We agree that clinical data is crucial to demonstrate the poor prognosis of basal-like tumors. For this, we have tried our best to collect the survival information of our cohort, and some patients' clinical data were collated (14 basal, and 18 classical). We have carried out the survival analysis to compare the difference between basal and classical tumors (Show below). Due to the incomplete data on our cohort, the Kaplan-Meier survival analysis displayed a p value of 0.18 which is not significant (determined by log-rank test). Despite this, we observed a clear tendency that the basal-like tumors exhibited shorter survival time than classical tumors. The worse survival of basal-like tumors have been demonstrated by several publications (e.g. Moffit et al., show below). Together, these results can prove the poor prognosis of basal-like tumors. Considering that p value is not significant, these results are not presented in the manuscript. We believe that a much larger cohort of PDAC will display a significant difference in survival time.

2) A major concern of the conducted microbiome experiments is the complete lack of any negative controls. Neither non tumoral-tissue (e.g. adjacent duodenum) nor elution or library preparation buffers are used to account for likely contaminations that occur in almost any of these microbiome experiments. To this end, a recent paper published by Nejman et al. (PMID: 32467386) sets a need standard regarding the required controls. The reported significant associations could otherwise be purely coincident findings without meaningful biological implications

Response: As suggested by the reviewer, we have supplemented several experiments and included negative controls to account for possible contaminations. Following the experimental design by Nejman et al., we introduced several types of controls including adjacent tissues, environmental control, DNA extraction control, and PCR no-template control to detect contaminations (show in Figure S4). The data demonstrated the presence of bacteria within PDAC tumors and proved that they are not originated from contaminations.

We have added the contents to manuscript on Result (page 7, line 208-215), Method (page

18, line 504-512) and Supplementary Figures (page 23).

3) The displayed genus in Figure 2b are predominantly present in the environment that occur in water, paper mill waste (e.g. *Dechloromonas*) and freshwater in Korea (*Sphingopyxis*). *Sphingopyxis* has never been shown to be present in human organisms. This again requires negative controls from samples stored in the same freezer to avoid likely cross-contamination. The absolute requirement of including negative controls in any microbiome analysis is nicely summarized in Rob Knight et al. (Nature Reviews Microbiology 2018).

Response: The concern about the contaminations in microbiome study is important. For this, we introduced several controls using PCR assay as described in Method. We focus at three crucial bacteria (*Acinetobacter*, *Pseudomonas* and *Sphingopyxis*) which were identified to be highly associated with PDAC progression by our analysis. The results confirmed the presence of these bacteria in PDAC and largely eliminated the probability of contaminations (show in Figure S4). Therefore, we believe that the main findings of this study are reliable.

Besides, we have compared the microbial communities of PDAC in our study to other publications (Riquelme et al. and Pushalkar et al.), and found many consistent results. The predominant bacteria such as *Pseudomonas*, *Elizabethkingia*, *Acinetobacter*, *Delftia*, *Agrobacterium*, *Sphingomonas* were also detected in PDAC patients by Riquelme et al. or Pushalkar et al. studies.

It should be noted that all samples in our study were processed with the same condition (same hospital, extraction protocol, library construction, sequencing platform), and this lead to a uniform effect if there is contamination. Therefore, these uniform contaminations will show no significant differences in abundance between subgroups, neither for association analysis.

4) The real presence of the top bacteria in basal like tumors should be confirmed by independent methods such as PCR, FISH or IHC.

Response: We have added the suggested experiments to the manuscript in Result (page 23). FISH was conducted using a specific probe against bacterial 16S rRNA sequences in FFPE PDAC samples. The data confirmed the presence of bacteria within PDAC tumors and demonstrated an increased abundance of bacteria in basal-like tumors compared with classical/hybrid subtypes (show in FigureS2). We performed IHC using antibodies against LPS to detect gram-negative bacteria in PDAC, and observed the consistent presence of bacterial LPS in our FFPE PDAC samples (Figure S3). PCR assays were executed to detect the candidate bacteria in PDAC tumors (Figure S4). Together, the presence of bacteria, especially the basal-related bacteria, was confirmed in our study.

5) As metagenomics sequencing is conducted, the author should attempt to analyze species not genera. The explanation in lines 169-171 is therefore not sufficient.

Response: We agree that analysis at species level would be more meaningful. However, we found several problems in analyzing species. Notably, there is a particular challenge to process accurate taxonomic assignment at species level due to the similarities among the individual genomes that belong to the same genus. For example, we found four species that belong to genus *Elizabethkingia* (*E. miricola*, *E. bruuniana*, *E. anopheles*, *E. ursingii*). Kraken2 classified only 250 reads to *E. miricola*, 170 reads to *E. bruuniana*, 55 reads to *E. anopheles*, 13 reads to *E. ursingii* at species level, while about 5,000 reads were classified to genus *Elizabethkingia*. Despite Bracken reallocated these reads from higher-level nodes to yield species abundance estimates, we found that reads were just re-assigned to these four species by certain proportion and the abundance estimates at species level were relatively rough. Therefore, we chose the genus level to perform the further analysis that will be more reliable. Besides, the raw data and the microbial abundance profiles at different taxonomic levels are available on NCBI and supplementary files that can be further analyzed with better approach.

Minor:

1) Line 50: wrong citation, no 13 fold increased risk is reported in this paper

Response: Thanks for your carefully checks. We feel sorry for our carelessness. This citation has been corrected.

2) Line 67: not Erick et al. but Riquelme et al (see citation 18)

Response: This mistake has been corrected.

3) Figure 2c: wrong statistical test: Anova + post-hoc tukey should be used here (given that normal distribution and homogeneity of variance is present)

Response: Thank you for pointing this out. We have re-analyzed the diversity measures using Anova with post-hoc Tukey HSD test. The legend and description of figure 2c have been revised accordingly.

>4) Figure 3c-e: y axis: which count is meant: read count? This depends of factores such as DNA input, Please show relative abundance

Response: In this study, the read count for each taxon was firstly measured by Kraken2+Braken, and the relative abundance was then calculated. The relative abundance is compositional data, and they are appropriate for diversity measurement or compositional presentation. However, standard statistical methods such as pearson correlation, t-text, Wilcoxon test, and so on are not directly applicable for analyzing microbial relative abundance data, which will cause inflated FDR ("*Aitchison, J. The statistical analysis of compositional data.*", "*Weiss, S. et al. Normalization and microbial differential abundance strategies depend upon data characteristics.*").

The read count (usually should be normalized) can be recognized as one type of absolute abundance. It is important to distinguish between absolute abundance and relative abundance. For example, suppose that Ecosystem A has 100 bacterial cells and 20 of them are *E. coli*, while Ecosystem B has 500 bacterial cells and 100 of them are *E. coli*. It is then reasonable to estimate that the relative abundance of *E. coli* is 20% in both Ecosystem A

and B. But the truth is there are more E. coli in Ecosystem B compared with A. Of course, the absolute abundance still has many problems such as sampling fraction. For this, we had controlled the consistent DNA input and sequencing size, and performed normalization to reduce the bias between samples. The differential abundance analysis was carried out using normalized absolute abundance, and they are shown in the y axis of Figure 3c.

Reviewer #2 (Remarks to the Author):

This is a well written manuscript that would be a welcomed and major contribution linking the fields of tumor genomic heterogeneity with microbiome.

The data is very nicely presented and the text is clear.

My only criticism is that the authors need to temper their language implying a cause and effect relationship between the microbial and genomic findings or vice versa. There is not enough information to imply causality. Regardless, this is a major paper that will be highly cited. The human dataset is impressive and builds on an important body of literature.

Reply to Reviewer #2

Dear Reviewer,

We feel great thanks for your professional review work on our manuscript. We really appreciate your positive comments. According to the suggestion, we have made several modifications to the description about the relationship between microbiome and host genomics in Result (page 11-12) and Discussion (page 15) sections. In this revised version, changes to our manuscript are highlighted by using red-colored text.

Reviewer #3 (Remarks to the Author):

In “Tumor microbiome contributes to an aggressive phenotype in the basal-like subtype of pancreatic cancer” Guo et al associate distinct subtypes of pancreatic cancer with their unique intratumoral microbial signatures. The manuscript is well written, premise is solid and investigates important interactions between tumor microbiota and tumor biology. I have the following questions:

Reply to Reviewer #3

Dear Reviewer,

We sincerely thank your valuable feedback on our manuscript. We have supplemented several data and made extensive modifications to the previous draft. Based on your comments, we attached a point-by-point responses as listed below.

1. The “mycobiome” has recently been shown to modulate pancreatic cancer (Aykut et al, 2019). Did the authors check for any fungus in the tumors?

Response: The publication by Aykut et al. presented a meaningful work and proposed the crucial role of fungal mycobiome in pancreatic cancer. They found that fungus *Malassezia* is able to promote PDAC. In our study, we did not detect this fungus by sequencing data. While, we had identified *Fusarium*, *Candida*, *Ustilago*, and *Aspergillus* in our PDAC samples, which were also found in pancreatic samples by Aykut et al. It should be noted, our analysis indicated that these fungi have very low abundance (much lower than bacteria), and they were only detected in particular cases. The abundance of microbiome (including bacteria, virus and fungi) were presented in Supplementary files (Table S1).

2. How did the authors control for laboratory contamination while looking for microbes inside the tumors? Did they use any negative controls for example, adjacent normal tissues?

Response: Thank you for your valuable suggestions to consider the contamination in microbiome study. To address this, we supplemented several experiments and introduced negative controls including adjacent tissues, environment control, DNA extraction control and PCR no-template control in PCR assay (Figure S4). The data largely proved that the candidate bacteria identified by our analysis were not laboratory-borne contaminations. We also compared the microbial communities of PDAC in our cohort to other publications, and found that the predominant microorganisms such as *Pseudomonas*, *Elizabethkingia*, *Acinetobacter*, *Delftia*, *Agrobacterium*, *Sphingomonas* were also detected by Riquelme et al. and Pushalkar et al. studies.

3. What were the baseline characteristics of subjects? Did they have a recent history of antibiotic intake or instrumentation as these can change the intrapancreatic microbial burden within various subtypes?

Response: Although patients display the relatively different cancer progressions, we tried to keep a consistent baseline characteristics of subjects. The eligible subjects in our cohort required a radiologic or histologic diagnosis of PDAC with resectable tumor. Patients who had received antibiotic treatment within past one month were excluded. Some pathologic information of patients had been collected, and no symptoms related to infection were observed. We presented a brief description about the subjects' recruitment in Methods (page 16).

4. Many dead bacterial products may be carried to the pancreas by the phagocytes and sequencing results may not be able to differentiate dead microbial DNA from that of live microbes. Did the authors try to culture pancreatic tumors to see if basal-like subtype indeed has increased growth compared to other subtypes?

Response: Dear viewer raised a key issue that metagenomics sequencing can't distinguish dead or alive bacteria cell or even the fragments of bacterial DNA molecules. To answer this question, we have performed FISH to detect the presence of bacterial 16S rRNA sequences (Figure S2). The immunohistochemistry (IHC) was also conducted using antibodies against LPS to detect gram-negative bacteria in PDAC tumors (Figure S3). Our

results confirmed the presence of intratumor bacteria within PDAC. As for bacteria culture, we feel quite difficult to process it due to the unknown culture condition for microbial communities. Our data demonstrated an increased abundance of bacteria in basal-like tumors compared with other subtypes.

5. Please mention other important works dealing with pancreatic cancer and the microbiome (Aykut et al, 2019; Sethi et al, 2018; Nejman et al, 2020). How did the author's findings compare to the tumor microbiota presented in Nejman et al and Riquelme et al ?

Response: We appreciate for your great suggestions, we have made changes in Introduction and Discussion parts to summarize these important researches regarding microbiome in pancreatic cancer. The comparison for microbiome compositions with other publications have been added in Discussion.

"Pushalkar et al. reported a 1000-fold increase of intrapancreatic bacteria in PDAC and demonstrated that the tumor microbiome promotes oncogenesis by inducing immune suppression [17]. Moreover, Riquelme et al. demonstrated that tumor microbiome composition is able to influence pancreatic cancer outcomes by activating antitumor immune response [18]. The intratumor bacteria in pancreatic cancer were also confirmed by Nejman et al., and they found that bacteria are mostly intracellular and are present in both cancer and immune cells [19]. Besides, the fungal mycobiome was showed to be implicated in the pathogenesis of PDAC via activation of MBL by Aykut et al. study [20]. The essential role of the microbiome in immune regulation and oncogenesis was highlighted in recent years that modulations of the microbiome can be profound for the immunotherapy against pancreatic cancer [21]."

"Notably, the increased abundance of Pseudomonas was also observed in PDAC patients with short-term survival by Riquelme et al. study, which confirmed the protumorigenic effect of Pseudomonas as revealed in our study [18]. Besides, Pushalkar et al. identified Pseudomonas as the most abundant bacterial genus that translocated from gut to pancreas in PDAC patients [17]. As demonstrated by the comparison of pancreatic microbiome between PDAC and wild-type mice, higher level of Acinetobacter was observed in PDAC mice. The presences of predominant genera such as Elizabethkingia, Delftia, Agrobacterium, and Sphingomonas by our analysis were also detected in PDAC patients by previous studies [17,18]."

Reviewers' comments:

Reviewer #1 (Remarks to the Author):

Overall, my initial requests have not been sufficiently addressed, in particular the lack of species analysis and the lack of robust clinical data is not acceptable to draw meaningful conclusions. Furthermore, although some negative controls have now been included, the majority of experiments are not sufficiently controlled. I vote "reject", and I believe the other reviewer comments are too superficial. However, if you feel to publish the preliminary data, it might still be an interesting manuscript for the community

Reviewer #3 (Remarks to the Author):

Thank you for your response

Reviewers' comments:

Reviewer #1 (Remarks to the Author):

Overall, my initial requests have not been sufficiently addressed, in particular the lack of species analysis and the lack of robust clinical data is not acceptable to draw meaningful conclusions. Furthermore, although some negative controls have now been included, the majority of experiments are not sufficiently controlled.

Reply to Reviewer #1

We have tried our best to revise the manuscript according to the three major requests raised by the reviewer. We have included the species analysis and supplemented the clinical data for a meaningful conclusion. An extensive revision had been made which marked in red in our manuscript.

1. In our previous manuscript, we provided a brief explanation that we focused at genus level for a higher accuracy. In some cases, the actual assignment results turned out that most of the microbial reads were only classified to the upper nodes of taxonomy tree, and this caused an inaccurate estimate at species level. In current revised manuscript, to explore the key tumor microbiome at species level, we expanded selected key genera to species on the basis of the genus-level microorganisms identified by statistical analysis. This strategy can avoid the bias from inaccurate abundance of species. We have identified many key species, for example, *Acinetobacter pittii* and *Acinetobacter junii* as the representative species belonging to *Acinetobacter*, and these species also displayed significant enrichment in basal-like tumors. The association analysis was also conducted between the species and host. We have added many contents to describe the analysis results at species level (page 6,10,12, Figure S2,S3,S8,S9).

2. We have tried our best to collect the survival information of our cohort, and some patients' clinical data were collated. The survival analysis between basal-like and classical subtypes of PDAC was presented in Supplementary figure1. We have added an objective description about the outcomes between subtypes in Results (page 3). In addition, we have cited two published studies (Moffit et al., Chan-Seng-Yue et al.) to demonstrated the poor prognosis of basal-like tumors. Based on the gene expression analysis and immune cells infiltration by our study, we proposed that basal-like tumors are accompanied by activated cancer-related pathways and inflammation, thereby indicating a more aggressive phenotype in basal-like tumors. These conclusions were not overstated. We also performed the survival analysis depending on the abundance of certain microbial genera and species, and the significant results suggested that the specific tumor microbiome is of predictive value for PDAC outcomes. (page 6, Figure3, Figure S4)

3. We had carefully followed the suggestions by previous comments that we introduced adjacent tissues, DNA extraction buffers, samples from the same freezer and PCR controls to detect contaminations. The reviewer suggested us to refer to the published study by Nejman et al., which mentioned diverse controls including DNA extraction controls, no-

template PCR controls, sequencing run controls, empty paraffin controls, hospital and laboratory environmental controls because that they collected samples from several centers and hospitals. In this study, we had included sufficient controls as much as we can, and additional controls are not necessary based on following reasons: 1. Technically, all sequencing processes were conducted by a third-part company, and the library preparation controls and sequencing run controls are not available. 2. We used fresh tumors in this study, hence the empty paraffin controls are not applicable. 3. All tumor specimens were collected from the same hospital, and were processed with the consistent procedure (extraction protocol, library construction, sequencing platform in one batch in one company). Theoretically, there is a prevalent presence of microbiome at species or genus level rather than significant difference if with contaminations. Our findings of differentially abundant microorganisms by statistic analysis or association analysis are not influenced by these contaminations.

We hope our revision and explanation can address your concerns. Thanks for your valuable comments.